# Design of Flexible Pressure Sensor Based on Conical Microstructure PDMS-Bilayer Graphene

**DOI:** 10.3390/s21010289

**Published:** 2021-01-04

**Authors:** Lixia Cheng, Renxin Wang, Xiaojian Hao, Guochang Liu

**Affiliations:** 1State Key Laboratory of Dynamic Testing Technology, North University of China, Taiyuan 030051, China; b1706004@st.nuc.edu.cn (L.C.); haoxiaojian@nuc.edu.cn (X.H.); b200611@st.nuc.edu.cn (G.L.); 2Taiyuan Institute of Technology, Taiyuan 030051, China; 3Science and Technology on Sonar Laboratory, Hangzhou 310000, China

**Keywords:** conical microstructure, PDMS, bilayer graphene, flexible pressure sensor

## Abstract

As a new material, graphene shows excellent properties in mechanics, electricity, optics, and so on, which makes it widely concerned by people. At present, it is difficult for graphene pressure sensor to meet both high sensitivity and large pressure detection range at the same time. Therefore, it is highly desirable to produce flexible pressure sensors with sufficient sensitivity in a wide working range and with simple process. Herein, a relatively high flexible pressure sensor based on piezoresistivity is presented by combining the conical microstructure polydimethylsiloxane (PDMS) with bilayer graphene together. The piezoresistive material (bilayer graphene) attached to the flexible substrate can convert the local deformation caused by the vertical force into the change of resistance. Results show that the pressure sensor based on conical microstructure PDMS-bilayer graphene can operate at a pressure range of 20 kPa while maintaining a sensitivity of 0.122 ± 0.002 kPa^−1^ (0–5 kPa) and 0.077 ± 0.002 kPa^−1^ (5–20 kPa), respectively. The response time of the sensor is about 70 ms. In addition to the high sensitivity of the pressure sensor, it also has excellent reproducibility at different pressure and temperature. The pressure sensor based on conical microstructure PDMS-bilayer graphene can sense the motion of joint well when the index finger is bent, which makes it possible to be applied in electronic skin, flexible electronic devices, and other fields.

## 1. Introduction

With the development of materials science, sensors with flexible substrates have gradually attracted people’s attention, especially the enthusiasm for research on electronic skin is increasing. Due to the flexibility, high sensitivity, high fit, and comfort of electronic skin [1,2,3,4], it can sense different external pressure like human skin as a biomedical sensor, that is, it has smooth conductive tactile signals. Flexible sensors can be applied not only to the medical field, but also to wearable devices and intelligent robot systems [5,6,7]. Recently, the Zhenan Bao group in Stanford University developed a sensor that consists of a strain sensor coupled to a pressure sensor and the sensor is capable of classifying compliance of materials with high sensitivity, and it can also identify materials [8]. The Takao Someya group from University of Tokyo has demonstrated ultraflexible and conformable optoelectronic skins that introduce multiple electronic functionalities such as sensing and displaying on the surface of human skin [9]. Zheng Yan’s team at the University of Missouri has developed a flexible electronic device based on pencil and paper that can monitor a series of important biological signals of the human body in real time, such as skin temperature, electrocardiogram, instantaneous heart rate, etc. It can also analyze in situ three sweat markers (pH, uric acid, glucose) [10]. Chen et al. have developed a flexible pressure device with high sensitivity, the device consists of sandwiched thin paper covered with ultrathin gold nanowires between two pieces of polydimethylsiloxane (PDMS), which can sense different forces [11]. Stefan C.B. Mannsfeld et al. have proposed an organic thin-film pressure sensing structure, in which one of the key layers of the OFET structure is dielectric medium. It is composed of thin rubber with regular structure, and it has excellent pressure sensitivity [12]. This resistance change of the sensitive rubber film is also adopted in various sensing devices [13,14,15,16]. Many organic and inorganic materials (such as carbon nanotube film [17,18,19,20,21] and graphene [22,23,24,25]) are now used to make various flexible electronic devices, pressure sensors, or stress sensors, and have better sensitivity. The mechanical excitation of these flexible sensors includes pressure, strain, shear, and vibration, etc. The basic conversion mechanisms for sensing mechanical quantities are piezoresistive, capacitive [26,27], piezoelectric [28], and so on. These mechanisms usually have better performance, and the relevant information of sensor is shown in Table 1 [29,30,31,32,33,34].

As an emerging two-dimensional single-layer carbon atom structure, graphene has excellent mechanical, thermal, optical, and electrical properties, and can realize different types of new micro-nanoelectronic devices. The pressure sensor made by Smith A.D. et al. with single-layer graphene has the detection range of 100 kPa, but the sensitivity is 2.25 × 10^−3^ kPa^−1^ [28]. The sensitivity of the pressure sensor made by Yao H.B. et al. with graphene mixed foam is 0.26 kPa^−1^. When the pressure is greater than 2 kPa, the sensitivity drops to 0.03 kPa^−1^ [31]. J. Zhang et al. have made a flexible pressure sensor using a layer-by-layer assembling reduced graphene oxide based on the micropyramid PDMS, which can reach the sensitivity of −1.71 kPa^−1^ in the range below 2 kPa. When the pressure increases above 2 kPa, the sensitivity drops to −0.02 kPa^−1^. As the pressure increases, the sensitivity further decreases [32]. It is difficult for graphene pressure sensor to meet both high sensitivity and large pressure detection range at the same time. Thus, it is of great significance to study the graphene pressure sensor with high sensitivity in a wider pressure range. The common flexible substrate of the flexible pressure sensor is plane structure, pyramid shape, etc. However, the manufacturing costs of some flexible sensors are relatively high while the process is complicated, which limits the wide application of a flexible sensor. The pressure sensor based on piezoresistive effect has the advantages of simple structure and high sensitivity. In this paper, a method for manufacturing a flexible pressure sensor based on conical microstructure PDMS with bilayer graphene is proposed, and the PDMS substrate with a conical microstructure is made by processes such as lithography, deep reactive ion etching, and rolling-over, etc. First the bilayer graphene (1 × 1 cm) was transferred to the flat PDMS substrate, then the electrodes were made on both sides of the graphene, then the PDMS (2 × 2 cm) substrate was covered with the conical microstructure on top of the graphene, and finally it was packaged as a flexible pressure sensor. The manufacturing process was relatively simple. After encapsulation, the normal-temperature probe station EPS150TRIAX, semiconductor analyzer 4200-SCS and the pressure testing machine (ZQ-32) were used to carry out relevant tests. The results show that the sensor with conical microstructure has higher sensitivity than that of flat PDMS when the applied force is the same. The working range of the bilayer graphene pressure sensor based on the conical microstructure PDMS is 0–20 kPa, and the sensitivity is 0.122 ± 0.002 kPa^−1^ (0–5 kPa) and 0.077 ± 0.002 kPa^−1^ (5–20 kPa), respectively. This flexible sensor has better sensitivity and reproducibility within a certain working range, and has certain reference significance for the application of flexible devices in other fields.

## 2. Finite Element Simulation of Flexible Pressure Sensor

When a vertical force is applied on the flexible substrate, the stress on the substrate will change. The piezoresistive material attached to the flexible substrate can convert the strain change caused by the vertical force into the change of resistance. In order to study the performance of the sensor, the finite element method (FEM) was adopted for modeling, which can determine the stress distribution on the flexible substrate after the vertical force is applied. Since the sensitivity of the sensor is closely related to the mechanical deformation after the vertical force is applied to the piezoresistive material, the structural deformation analysis was adopted to detect the local stress change of the piezoresistive material. In the simulation analysis, the piezoresistive material was bilayer graphene. The commercial software COMSOL Multiphysics (v.5.5) was used to perform related simulation. In the established FEM model, the length and width of the PDMS were 518 μm, the thickness of PDMS in the FEM model was 500 μm, the material parameters of PDMS in the model were reflected by Young’s modulus of 750 kPa and Poisson’s ratio of 0.49. The length and width of the bilayer graphene in the model was also 518 μm, and the bilayer graphene parameters used in the COMSOL model were obtained in literature, such as Young’s modulus of 1030 GPa and Poisson’s ratio of 0.16 [35,36]. The conical geometry used here was as following: the top diameter of this conical structure was 5 ± 2 μm, the bottom diameter was 79 ± 5 μm, the height was 98 ± 5 μm and the center distance between the adjacent two conical microstructures was 180 ± 5 μm. When vertical pressure was applied, the piezoresistive material on the flexible substrate was deformed. The bilayer graphene piezoresistive material was used to convert the local structural surface deformation of the substrate to the change of resistance. When simulating the substrate with conical microstructure, due to the small size, the calculation amount was too large when the simulation was performed after all modeling was completed, so nine of the conical microstructures were selected for the analysis. When 12 kPa pressure was applied vertically above the flexible sensor, the stress distributions obtained on the conical microstructure and flat substrate were shown in Figure 1a,b, respectively. It can be seen from Figure 1a that the stress can be achieved as 4.96 × 10^6^ Pa at the bottom of the conical structure on the contact surface under 12 kPa. We can see from Figure 1b that the max stress is only 1.23 × 10^4^ Pa in the center of the flat PDMS under 12 kPa. It can be clearly seen that under the same pressure, the stress on the contact surface of the conical microstructure is much larger than the one without this structure. In order to obtain the change of displacement of the same point under different pressures, three center points parallel to the OX axis and located on the same line were taken to observe its displacement under pressures of 0, 5, 10, 15, 20, and 25 kPa, respectively. The total displacements obtained on the conical microstructure and flat substrate are shown in Figure 1c,d, respectively. It can be observed that when the substrate is the same, the greater the pressure applied is, the greater the displacement becomes at the same point. When the same pressure is applied to the substrate with conical microstructure, the bottom of the three conical microstructures located on the same line will produce the largest displacement, which is also the first part to contact the piezoresistive material, and the displacement of other positions is significantly reduced. Under the same pressure, the displacement at the bottom of the conical microstructure substrate is more than five times than that of the same point on the flat substrate. Therefore, it can be seen from the simulation results that under the same conditions, the substrate with conical microstructure produces a larger amount of deformation and resistance change, which is more suitable for flexible pressure sensors.

The influence of the number of conical microstructures on the stress and deformation was studied by the simulation software. After setting up the relevant model in the simulation software COMSOL, each size of the conical microstructure was certain, then the distance between adjacent two conical microstructure was set as variable, after parametric scanning for this variable, the stress diagram of the vertex of each conical microstructure under the same pressure and stressed area was obtained. It can be seen from the established model that the number of conical microstructures in the same stressed area was reducing from 36 to 4 with the distance between the adjacent two conical microstructures increasing from 80 to 250 μm. The variation stress at the vertex of the conical microstructure with the distance between the adjacent two conical microstructures is shown in Figure 2. When the same pressure is applied, the stress at the vertex of each conical microstructure will increase with the number of conical microstructures decreasing, so with the greater deformation of the material. However, with the increasing of the stress, the critical load of the stress surface will be gradually reached. After buckling analysis of the model with different numbers of the conical microstructures, we find that the critical load of the stress surface is about 5 × 10^6^ N/m^2^, if more than the value, the material of the stress surface will be damaged. When the value is 5 × 10^6^ N/m^2^, the corresponding distance between the adjacent two conical microstructures is 180 μm in Figure 2. Finally, we determined the optimal size. When the distance between the adjacent two conical microstructures is 180 μm, the number of the conical microstructure is nine.

## 3. Materials and Methods

### 3.1. The Preparation of Flexible Pressure Sensor

Due to the unique single-layer atomic structure of graphene, the large-area graphene cannot support its structure. In order to maintain the required shape, it is necessary to transfer graphene to the substrate. Graphene is adsorbed onto the substrate surface by van der Waals forces to make a high-sensitivity flexible sensor. PDMS is a polymer organic silicon compound, optically transparent, nontoxic, nonflammable, and has good adhesiveness to silicon slices, it also has excellent flexibility. Hence, PDMS was selected as the substrate of graphene, which was the main contact surface of the flexible pressure sensor.

The flexible pressure sensor was prepared during a series of processes, and the specific process flow chart is shown in Figure 3. The specific process steps corresponding to Figure 3 are as follows: (a) spin-coating of photoresist. After cleaning the silicon slice, the 4-inch silicon slice was put into a vessel containing acetone and absolute ethanol, then the glass rod was used to press it, and ultrasonic cleaning was performed for 20 min. Then it was rinsed with deionized water and dried with N_2_ gun to ensure the clean surface of the slice. Surface treatment was carried out on the silicon slice. It was coated with HMDS, and placed in a vacuum oven at a temperature of 130 °C to prevent degumming after development. Spin-coating of AZ6130 photoresist was performed at 500 r/min for 5 s, then 3000 r/min for 20 s, and the thickness of the glue was about 3 μm. Then it was prebaked for 90 s at 100 °C; (b) lithography. EVG’s 610TB contact lithography machine was used to expose at an exposure dose of 150 mJ/cm^2^, then developed with AZ238 developer solution for 30 s, and then placed on a drying table at a temperature of 120 °C to bake for 3 min; (c) deep reactive ion etching (DRIE). SPTS’s LE0765LPX DSI deep silicon etching machine was used for etching, the pre-etching depth was 100 μm and the pre-etching angle was 90°. The temperature was 20 °C, the depth to width ratio was 50:1, and the etching rate was 1 μm/loop; (d) plasma stripping. The IoN Wave10 plasma stripper from PVA TePla AG was used to remove the adhesive. The relevant parameters were set as follows: O_2_ flow rate was 3 L/min, power was 500 W, and time was 3 min; (e) depositing Parylene after wet etching. During the wet etching process, the HNA etching was configured. The etching solution was made up of HF, HNO_3_, HAC with the mass fractions of 40%, 65%, and 100%, respectively, in a volume ratio of 1:8:3. Nitric acid, hydrofluoric acid, and glacial acetic acid were mixed in a certain proportion and stirred well. After corrosion, a conical microstructure was obtained. After cleaning, it was put into the PDS2010 Parylene vacuum vapor deposition apparatus of SCS company, and a layer of Parylene film with a thickness of 300 nm was deposited; (f) pouring the prepared PDMS. The Sylgard184 PDMS A was selected. PDMS and curing agent were well mixed in 10:1. Air bubbles were drawn out in the vacuum drying oven, and the mixture was evenly poured on the silicon slice to ensure that the position where the silicon slice was placed was flat; (g) depositing Parylene after tearing off PDMS. It was left there for 4 h. The thickness of liquid was uniform on the silicon slice. After the stepwise heating and curing (the temperature was increased in the five temperature stages of 30, 40, 50, 60, and 70 °C, respectively. The first four stages lasted for 5 min, and the last stage continued for 2.5 h), the tweezers were used to tear off the PDMS from the silicon slice slowly, and then a 300 nm thick Parylene film was deposited on the side with conical microstructure; (h) pouring the configured PDMS again. Then the same heating and curing process was carried out; (i) tearing off PDMS. PDMS was teared off slowly, and then the microstructure PDMS was cut into 2 × 2 cm blocks; (j) transferring graphene. The bilayer graphene was purchased from Hefei microcrystalline materials technology company. The size of the bilayer graphene was 1 × 1 cm. As the surface of bilayer graphene was coated with a PMMA protective film, it was necessary to remove the PMMA after transferring the bilayer graphene to the target substrate. First 500 nm thick Parylene was deposited on the flat PDMS to enhance the adhesiveness to the metal, and then it was cut into 2 × 2 cm blocks. The bilayer graphene coated with PMMA protective film in deionized water was released separately, a flat piece of PDMS was clamped with tweezers, and the graphene was transferred to the target substrate. Then it was dried at room temperature for 20 min, and then baked at 70 °C for 30 min. It was cooled to room temperature and soaked in acetone for 10 min, then transferred to the second box of acetone and soaked for 30 min. The purpose was to remove PMMA (considering that PDMS may be soluble in acetone, a piece of PDMS was soaked in acetone separately after placing it for 4 h. The PDMS only slightly swelled and there was no obvious change, so PMMA can be removed by soaking in acetone); (k) sputtering Ti and Au. The middle part of the graphene was covered with aluminum foil and fixed on a clean silicon slice with tape. It was put into a magnetron sputtering machine, then sputtering was undertaken at 20 nm Ti and 150 nm Au, aiming to enhance the adhesiveness of the conductive part and the flexible substrate; (l) encapsulating after coating conductive silver glue. After the sputtering, the aluminum foil was removed and the PDMS placed with the conical microstructure on top of the graphene. The two wires were placed on both sides of the graphene, and an appropriate amount of conductive silver glue was coated at the same time. Heating and curing was performed for 20 min, and it was encapsulated into a sandwich structure. When manufacturing the PDMS substrate without the microstructure, the above steps can be simplified.

### 3.2. Characterization of Flexible Pressure Sensor

In the preparation of the sensor, SUPRA-55 field emission scanning electron microscope (SEM) of ZEISS company was used for characterization. After the lithography, photoresist was used as a mask to perform dry etching, that is, deep reactive ion etching. It is a high-density plasma etching with the advantages of high control accuracy and less etching damage. The etching gas was C_4_F_8_ and SF_6_. The silicon pillar obtained after deep reactive ion etching had a bottom diameter of 80 ± 5 μm, a height of 100 ± 5 μm, and a center distance of 180 ± 5 μm. The silicon pillar array was relatively intact, but there were a few burrs on the edge of single silicon pillar. These burrs are mainly caused by the photoresist eating in the etching process, but it does not affect the overall silicon pillar array. In order to obtain the conical microstructure, the silicon pillar array needed to be wet-etched. After the HNA etching solution was configured, the etching was carried out after continuous stirring. After corrosion, the silicon column with conical microstructure was obtained, the top diameter of this microstructure was 5 ± 2 μm, the bottom diameter was 79 ± 5 μm, the height was 98 ± 5 μm, and the center distance of two adjacent conical microstructures was 180 ± 5 μm. The edge corrosion rate of the silicon pillars was relatively high due to the isotropic corrosion of silicon. Figure 4a shows the overall structure after corrosion measured by scanning electron microscope FESEM. In this way, the conical microstructure was formed. The SEM of a single column with conical microstructure is shown in Figure 4b.

Raman spectroscopy is an important means to characterize the graphene lattice. During the preparation process, Raman test was performed on the PDMS after transferring the bilayer graphene to it, and the test results are shown in Figure 5. It can be seen from Figure 5 that the green color shows the G band and the red one is the 2D band. The intensity values of the G and 2D peaks appear at 1582 and 2675 cm^−1^, respectively, and the D peak appears at 1346 cm^−1^. The negligible intensity value of D peak indicates that the bilayer graphene has good quality and fewer defects after the transfer. After the peak-differentiating and imitating, the ratio of the peak intensity of the 2D peak (2675 cm^−1^) to that of G peak (1582 cm^−1^) is 0.6 (I_2D_:I_G_ = 0.6), which is less than 1, indicating that the graphene is non-monolayer. The resistance of the bilayer graphene on PDMS substrate is 1.71 kΩ, which is slightly larger than the resistance of transferring it to the SiO_2_ substrate. The resistance of the bilayer graphene on the SiO_2_ substrate is 1.58 kΩ. This difference is attributed to the slight tensile strain after the transfer on the flexible substrate.

### 3.3. Test on Flexible Pressure Sensor

The flexible pressure sensor was placed on the lower pressure table of the pressure testing machine (ZQ-32), and the wires from both sides of the flexible pressure sensor were connected to the two ends of the EPS150TRIAX room-temperature probe station. The probe station was connected to the SUM1 and GNDU of semiconductor analyzer 4200-SCS, respectively. The pressure testing machine (ZQ-32) was used to apply different pressure vertically to the pressure sensor in order to test the pressure performance of the sensor. The test diagram is shown in Figure 6, and the physical test diagram is shown in Figure 7. The sensitivity S is defined as Equation (1), where R_0_ and R denote the resistance without and with applied pressure, respectively, and ΔP denotes the change of pressure.
(1)$$S=\frac{\left(R-\;R0\right)/R0}{\Delta \;P}$$

## 4. Results and Discussion

The sensitivity curves of flexible pressure sensor with and without conical microstructure PDMS are shown in Figure 8. It can be seen that the sensor with conical microstructure can operate in the pressure range of 0–20 kPa, and the sensitivity is 0.122 (0–5 kPa) and 0.077 kPa^−1^ (5–20 kPa), respectively, and the resistance remains almost unchanged after more than 20 kPa. The sensitivity of the sensor without conical microstructure PDMS is 0.072 (0–5 kPa) and 0.042 kPa^−1^ (5–20 kPa), respectively, in the range of 0–20 kPa. We can see that the pressure sensor with conical microstructure can operate at a pressure range of 20 kPa while maintaining a relatively high sensitivity. It can be seen that the sensitivity of the sensor with conical microstructure is higher than that without conical microstructure. It also can be seen that the sensitivity of the sensor in the working range of 0–5 kPa is higher than that of 5–20 kPa, regardless of whether the PDMS has a conical microstructure. As the thickness of the bilayer graphene is very thin, a slight pressure change will cause a significant change in resistance. However, when the applied pressure is greater than 5 kPa, the deformation of graphene is not as obvious as before, so the change of resistance gradually decreases, which demonstrates the sensitivity in the working range 5–20 kPa is lower than that of 0–5 kPa. What is more, when the pressure increases to more than 20 kPa, the deformation of the bilayer graphene is close to saturation, so the resistance no longer changes significantly. It can be seen that for bilayer graphene pressure sensor, the sensitivity of the pressure sensor with conical microstructure PDMS has been improved to a certain extent.

The reproducibility of sensor is an important indicator to measure the performance of the sensor. We performed pressure tests on sensors with conical microstructure PDMS. Five flexible sensors prepared by the same process were tested under the room temperature of 24 °C. The resistance responses of the sensors were tested by applying and releasing different pressures vertically above the sensor. Figure 9 shows the variation of resistance of each sensor when 5 and 15 kPa pressures were applied and released. As can be seen from the figure, after hundreds of tests, when the same pressure is applied to different sensors, the resistance variation range of the sensor is slightly changed, indicating that the process is well prepared, and the consistency and the repeatability of the sensor is relatively good.

Linearity is an important index to describe the static characteristics of a sensor. The origin software was used to conduct linear fitting for the testing data of the flexible pressure sensor, and the linearity curve of the flexible sensor is shown in Figure 10. The equation of the fitted line is y = 0.06714x + 0.17605. After calculation, the linearity of this curve is 6.92%, which indicates that the linearity of the sensor is relatively good.

In order to find the relationship between the performance of the sensor and the change of the temperature, we tested the rate of resistance change on the same sensor with the conical microstructure under different temperatures of 15, 24, and 33 °C, respectively. The sensitivity curves of the sensor are shown in Figure 11. It can be seen from the figure that the sensitivity of the sensor working at the room temperature 24 °C is a little higher than that of 15 and 33 °C when the applying pressure is within 5 kPa, but the difference is very small. As the performance of the materials is most stable in the room temperature, when the applying pressure is small, the change of resistance is the most obvious. When the applying pressure is increasing from 5 to 20 kPa, we can see from the figure that the sensitivity of the sensor is almost the same. It indicates that the sensor can work stably at different temperatures and has a relatively good reliability.

In order to find the response time of the flexible sensor, we enlarged the response characteristic figure of the sensor when a certain pressure was applied as shown in Figure 12. It can be seen from the figure that the change of resistance slightly lags behind the change of pressure, the lagging time to reach the stable state is the response time of the sensor. We can see the response time of the sensor is about 70 ms, indicating that the response time of the sensor is relatively short.

Since the sensor adopts the flexible substrate and has a better flexibility, the sensor can be fixed to a certain part of the human body to monitor related physiological signals. Due to the thinner sensor with bilayer graphene sandwich that fits better with the index finger joints, it was fixed to the joint of the index finger. As the joint bends, the resistance response performance is shown in Figure 13c. The output resistance of the sensor with bilayer graphene is almost consistent when the joint is bent to a certain degree. Figure 13a shows the state when the joint is bent, and Figure 13b shows the state when the joint is straight. It can be seen that this sensor can better sense joint movement and has a relatively stable output.

## 5. Conclusions

We proposed a high sensitivity flexible pressure sensor preparation based on conical microstructure PDMS-bilayer graphene through a relatively simple and low-cost process. The flexible pressure sensor based on conical microstructure PDMS showed a higher sensitivity than that of flat PDMS when the morphology of graphene and the applied force are the same. We found that the pressure sensor based on conical microstructure PDMS-bilayer graphene has a large working range (0–20 kPa) and relatively high sensitivity. The sensitivity of the flexible pressure sensor with microstructure PDMS-bilayer graphene is 0.122 ± 0.002 kPa^−1^ (0–5 kPa) and 0.077 ± 0.002 kPa^−1^ (5–20 kPa), respectively. Additionally, the sensor can work stably at different temperatures and the response time is about 70 ms. In addition to the high sensitivity, the cycling stability of the sensors was demonstrated to output repeatable and stable signals over hundreds of cycle tests. By attaching the flexible pressure sensor (bilayer graphene) to the index joint, the movement of the index joint can be measured in real time. The high sensitivity, stable performance, and low-cost fabrication of the pressure sensor make it a promising candidate for electronic skin and other aspects. As our sensors have relatively high sensitivity and flexibility, it can be attached to a part of the body to detect some physiological signals of the human body such as pulse, heart sound, and so on, in the future.

## Figures and Tables

**Figure 1 sensors-21-00289-f001:**
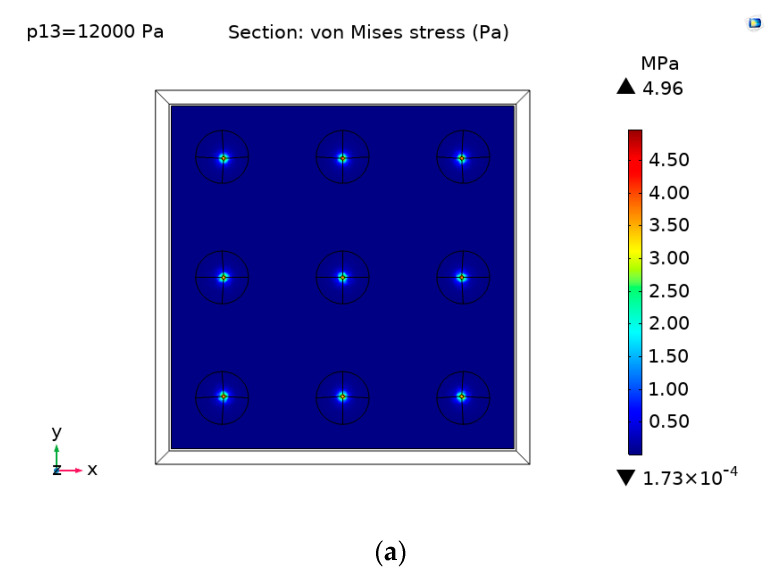
The von Mises stress induced by applying vertical force on the flexible sensor. (**a**) Distribution of the von Mises stress induced by applying a vertical force of 12 kPa on the flexible sensor with conical microstructure polydimethylsiloxane (PDMS). (**b**) Distribution of the von Mises stress induced by applying a vertical force of 12 kPa on the flexible sensor with flat PDMS. (**c**) The change of the displacement of three center points under different pressures (0, 5, 10, 15, 20, 25 kPa), the three center points are parallel to the OX axis and located on the same straight line on the substrate with conical microstructure. (**d**) The displacement change diagram by taking the same three center points on the flat substrate after applying 0, 5, 10, 15, 20, 25 kPa pressure, respectively.

**Figure 2 sensors-21-00289-f002:**
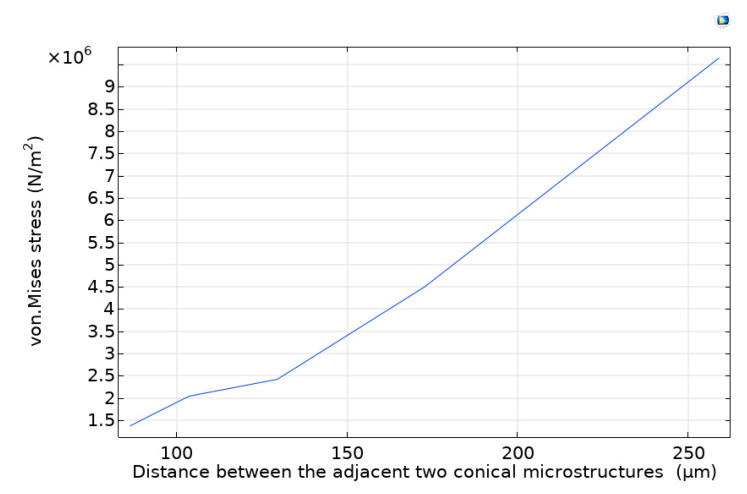
The diagram of variation stress at the vertex of the conical microstructure with the distance between the adjacent two conical microstructures.

**Figure 3 sensors-21-00289-f003:**
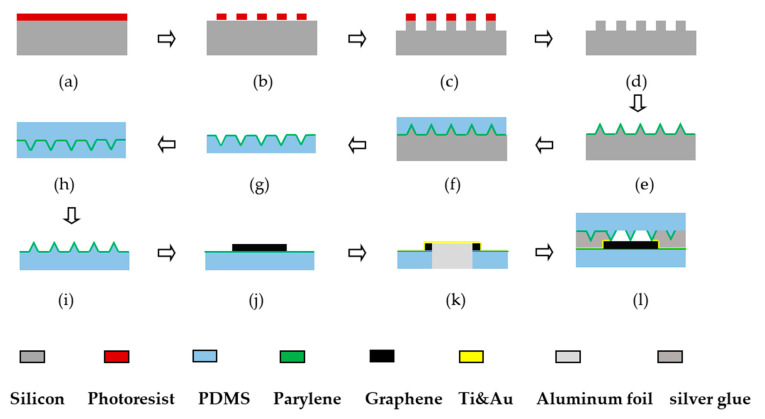
The flow chart for preparation process. (**a**) Spin-coating of photoresist. (**b**) Lithography. (**c**) Deep reactive ion etching. (**d**) Plasma stripping. (**e**) Depositing Parylene after wet etching. (**f**) Pouring the prepared PDMS. (**g**) Depositing Parylene after tearing off PDMS. (**h**) Pouring the configured PDMS again. (**i**) Tearing off PDMS. (**j**) Transferring graphene. (**k**) Sputtering Ti and Au. (**l**) Encapsulating after coating conductive silver glue.

**Figure 4 sensors-21-00289-f004:**
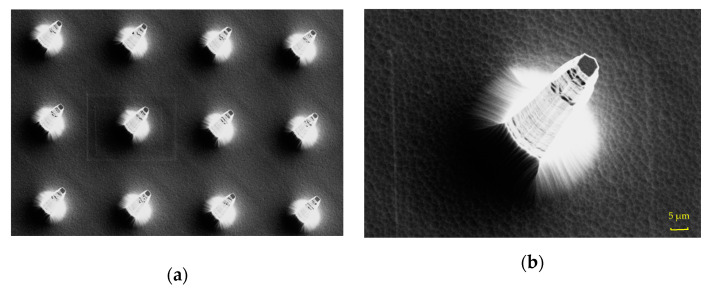
Characterization of conical microstructure after wet etching. (**a**) The morphology characterization diagram of silicon array with conical microstructure after wet etching; (**b**) the morphology characterization of single silicon column with conical microstructure. This top diameter of this structure is 5 μm, the bottom diameter is 79 μm, and the height is 98 μm.

**Figure 5 sensors-21-00289-f005:**
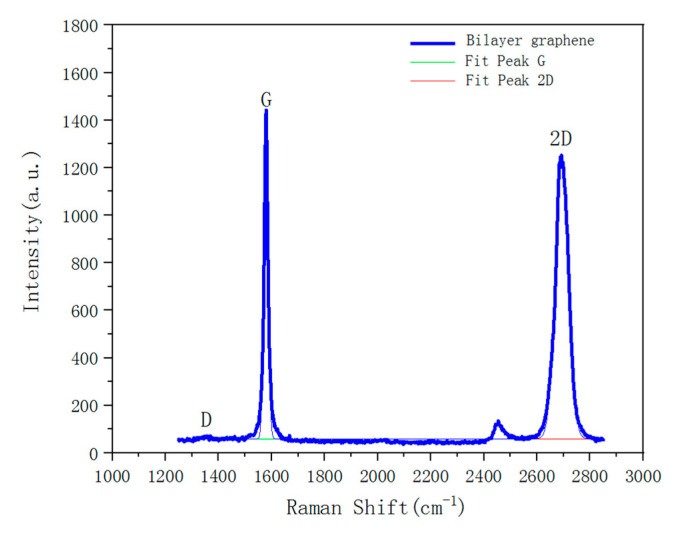
Raman frequency shift of bilayer graphene after transferring it to the PDMS substrate. The values of G peak and 2D peak appear at 1582 and 2675 cm^−1^, respectively. The ratio of the peak intensity of 2D peak and G peak is I_2D_:I_G_ = 0.6. The intensity of the D peak is relatively low after transfer, indicating that the graphene has fewer defects after the transfer.

**Figure 6 sensors-21-00289-f006:**
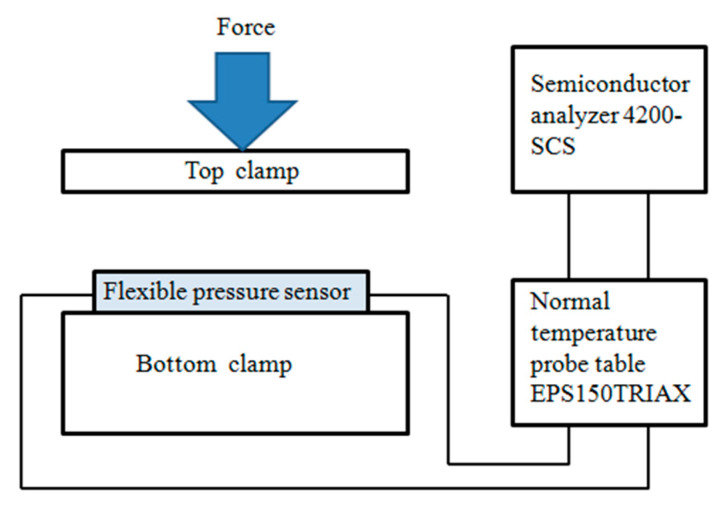
The diagram of testing system for flexible pressure sensor.

**Figure 7 sensors-21-00289-f007:**
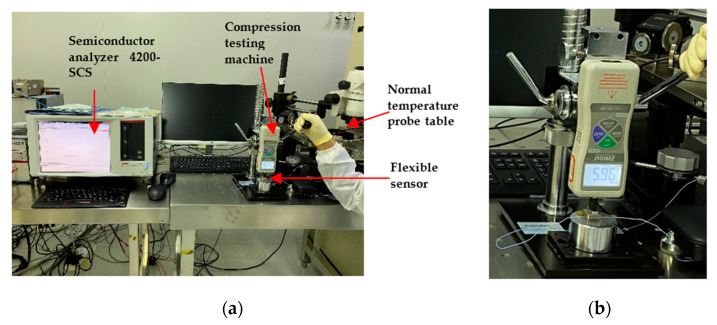
(**a**) The physical test diagram of flexible pressure sensor; (**b**) pressure testing machine ZQ-32.

**Figure 8 sensors-21-00289-f008:**
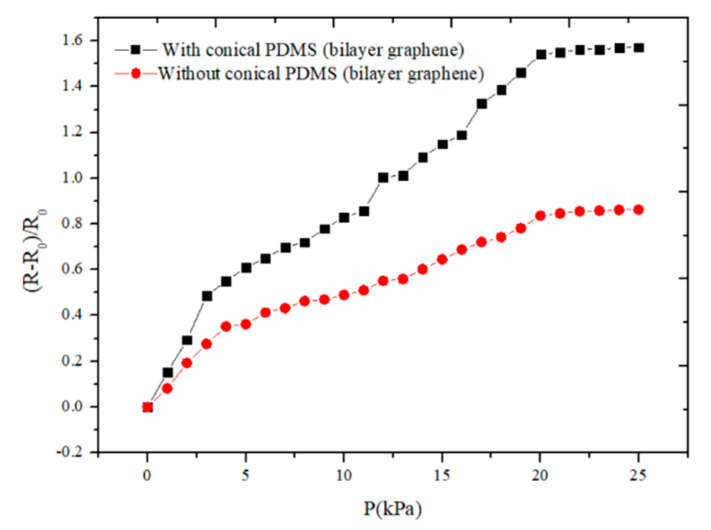
The sensitivity curves of the sensor with and without conical microstructure PDMS. The dimensions of the conical microstructures PDMS are as following: the bottom diameter of each conical microstructure is 79 ± 5 μm, the top diameter is 5 ± 2 μm and the height is 98 ± 5 μm, the center distance of two adjacent conical microstructures PDMS is 180 ± 5 μm.

**Figure 9 sensors-21-00289-f009:**
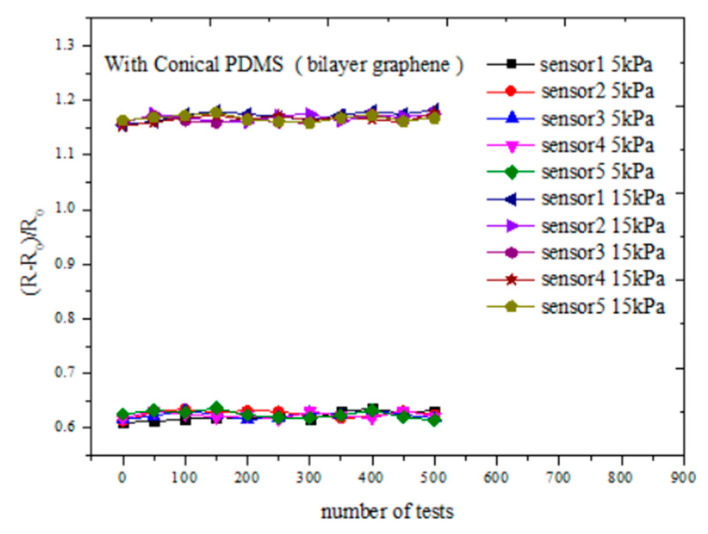
Sensor reproducibility test curves of conical microstructure PDMS (bilayer graphene) at 5 and 15 kPa, respectively. The dimensions of the conical microstructures PDMS are the same as in Figure 8.

**Figure 10 sensors-21-00289-f010:**
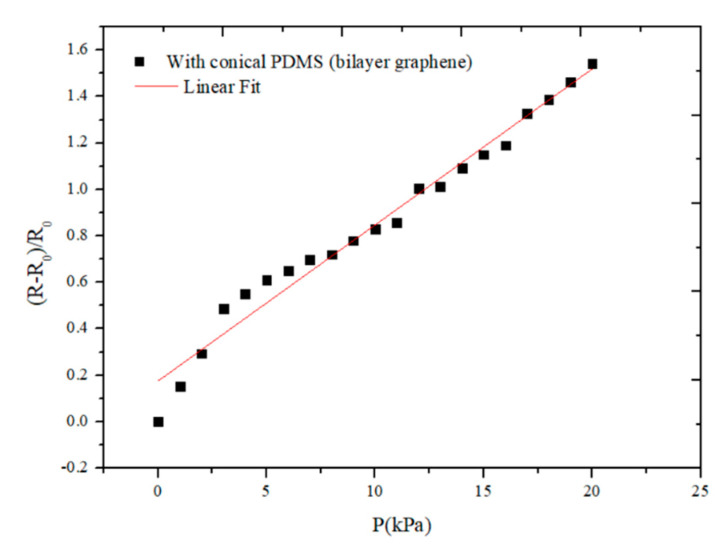
The linearity curve of the flexible sensor.

**Figure 11 sensors-21-00289-f011:**
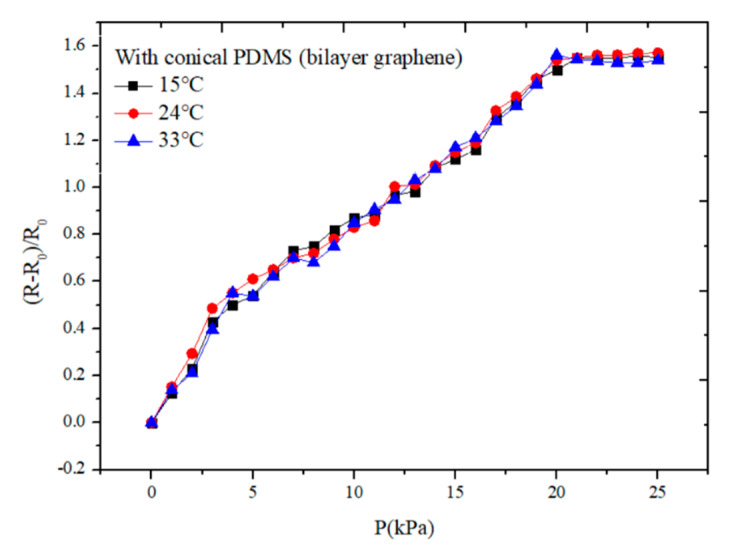
The sensitivity curves of the sensor with the bilayer graphene under different temperatures of 15, 24, and 32 °C, respectively.

**Figure 12 sensors-21-00289-f012:**
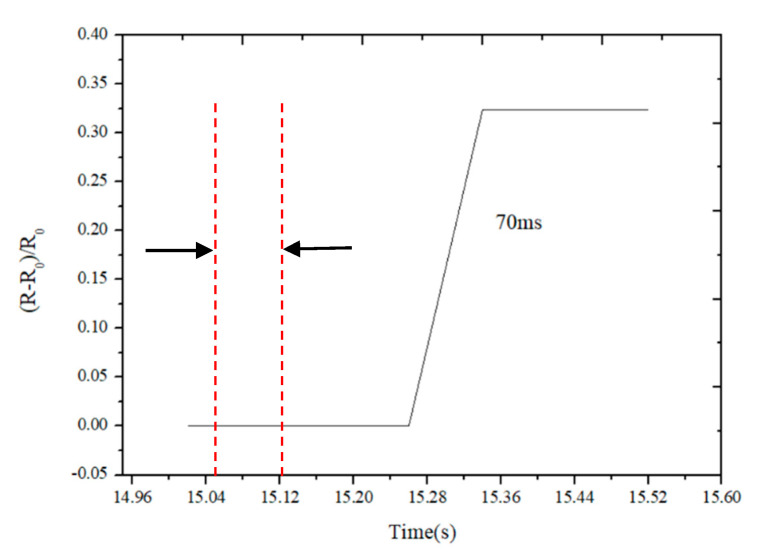
Response time of the flexible sensor.

**Figure 13 sensors-21-00289-f013:**
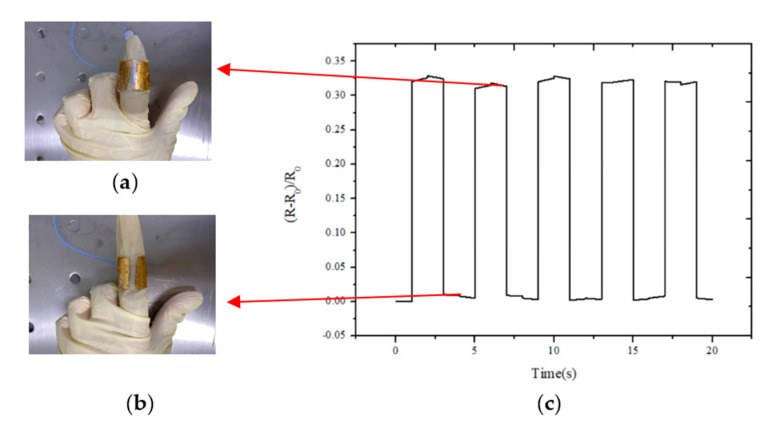
(**a**) Joint bending state; (**b**) joint straight state; (**c**) resistance output performance test of bilayer graphene sensor fixed at the finger joint.

**Table 1 sensors-21-00289-t001:** Summary of some flexible sensors and relevant performance parameters.

Source	Structural Figure	Substrate	Key Materials	Mechanical Component	Transduction Principles	Sensitivity (GF)	Range
S. Chun et al. [29]	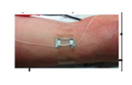	PET	Double-layered graphene	Pressure	Piezoresistivity	0.24 kPa^−1^ (<250 Pa) 0.039 kPa^−1^ (>700 Pa)	0.3 Pa–10 kPa
Smith A.D et al. [30]	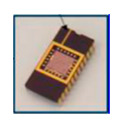	Cavities etched into a SiO_2_ film on a silicon substrate	Graphene membranes	Pressure	Piezoresistivity	2.25 × 10^−3^ kPa^−1^	0 Pa–100 kPa
Yao H.B et al. [31]	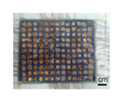	PDMS	Graphene-polyurethane spone	Pressure	Piezoresistivity	0.26 kPa^−1^ (<2 kPa) 0.03 kPa^−1^ (2–10 kPa)	0 Pa–10 kPa
J. Zhang. et al. [32]	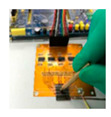	Micro- pyramid PDMS	Reduced graphene oxide (RGO)	Pressure	Piezoresistivity	−1.71 kPa^−1^ (<2 kPa) −0.02 kPa^−1^ (2–5 kPa)	0 Pa–5 kPa
Sungwoo Chun et al. [26]	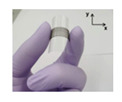	PEN	CNT sheets	Pressure	Capacitance	0.06–0.13% (<20 kPa) 0.02–0.04% (20–40 kPa)	1 Pa–40 kPa
Tran Quang Trung et al. [33]	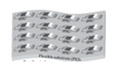	PES	Reduced graphene oxide FET	Strain	Piezoresistivity	0.02–0.35%	0–0.8%
S. Chun et al. [34]	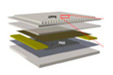	PEN	Single-layer graphene	Strain	Piezoresistivity	1.25–1.4%	24 Pa–3 kPa

## Data Availability

Data sharing not applicable. No new data were created or analyzed in this study. Data sharing is not applicable to this article.

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
