# Peer review of "Design of Flexible Pressure Sensor Based on Conical Microstructure PDMS-Bilayer Graphene"

_sensors, 2021, doi:10.3390/s21010289_

Round 1

Reviewer 1 Report

Cheng et al. describe the fabrication and characterization of a piezoresistive pressure sensor based on a composite of PDMS and graphene. The authors test two sensor architectures, i.e., one flat and one based on conical microstructures. They compare the relative variation of resistance and the total displacement for the two architectures in the 0-25 kPa range of pressures.

The draft must be improved because the English language has to be edited and figure 1 and figure 4 are very difficult to read.

In particular, Materials and methods section is extremely difficult to follow. I suggest to re-write it.

Other few minor revisions:

  • In the introduction section, when the authors describe electronic skin and their application, they should add references, in particular from Zhenan Bao group (Standford University) and Takao Someya group (University of Tokyo).
  • Page 1, line 36: please change PH with pH
  • The authors should report results and dimensions with the associated error; in particular, dimensions of the conical microstructures (page 6 line 207) and figure 7 and figure 8.
  • The author should report possible future applications of their sensors (applications in the 0-20 kPa range of pressure)
  • References must be reported all with the same format.

Author Response

Dear reviewer:

Reviewer 2 Report

Dear authors,

first many thanks for your interests to submit your reasearch results in our journals. The topic of graphene-based pressure sensor is quite interessting and hot at the moment and could be interessting for readers.

But after studying your paper I have concern to accept your paper. First it's quite thin and poor, and does not have enough background information for interessted readers. You have a quite good review part on published flexible pressure sensors using graphene.

1. The simulation part: it is to be improved, as it lacks many things:

-description of the FEM model used (parameters such as PDMS thickness, the sandwich structure).

-the conical geometry (size) used here?

-the influence of the number of conical microstructures on the stress and deformation. It's quite clear if you work with the same pressure you have very high force on the conical microtructure tips - and so high stress and deformation there.

-the results (graphs) have very low quality (scale, subscription etc.) have to be imroved. You've stated "It can be clearly seen 99 that under the same pressure, the stress on the contact surface of  the conical microstructure is much 100 larger than the one without this structure". Readers see nothing on your graphs.

2. Technology part:

-process flow is bad. The figure 2k is wrong

-wet etching solution - composition not clear

-no information about graphene layer

3. Characterization part:

-it laks a statistical analysis (how many sensors have been tested, the variation of conical sizes, number of conical microtips etc.)

-linearity of the sensor seems to be bad

-reliability not clear

-repeatability of the sensor signal not clear

-no explanatation of the two ranges (5k and 20k)

Best regards

Author Response

Dear reviewer:

      Please see the attachment .

      Thank you .

     Best wishes

Round 2

Reviewer 1 Report

The authors have improved the first version of the work, but I must ask again to modify

  • Figures, especially Figure 1 and Figure 2, since the axis are really difficult to read. Please rewrite axis and legend with a larger font.
  • Results and dimensions. They must be reported with the associated error. For instance, dimensions of the conical microstructure must be reported as "the top diameter of this structure is (5 ± XX) μm, the bottom diameter is (79± XX) μm, and the height is (98± XX) μm". Those are not absolute values, they have an intrinsic variability from one structure to another.

Author Response

Dear Reviewer: 

Reviewer 2 Report

Dear authors,

thanks again for your re-submission. Your paper still needs to be improved.

Please recheck your text again and improve formality. For examples:
Line 17 - Space
Line 20 - space kPa-1 - unities
Line 42 - Space
Line 99
Line 168
LIne 183
Line 188 change to HNA etching and not corrosion
Line 189 HNO3

Figure 4: please use better figures with scale. Readers might need more details of the silicon tips in the figures.

Part 4. Languags has to be improved
Reliability of your sensor has to be shown: Please give linearity in %, Temperature dependency and hystersis behaviour.

Best regards

HDN

Author Response

Dear Reviewer:
